

# Clinical characteristics, anti-SARS-CoV-2 IgG titers, and inflammatory markers in individuals with post-COVID-19 condition in Kenya: a cross-sectional study

Martin Theuri[1], Eric M. Ndombi[2], Peris Thamaini[3], James Opiyo Ogutu[2], Lister Onsongo[4], June K. Madete[5], Victor Ofula[6], Samuel Gitau[7], Gladys Mwangi[7] and Paul Okemo[8]

[1] Department of Medical Laboratory Science, Kenyatta University, Nairobi, Kenya
[2] Department of Medical Microbiology and Parasitology, Kenyatta University, Nairobi, Kenya
[3] Department of Human Pathology, Kenyatta University, Nairobi, Kenya
[4] Department of Community and Reproductive Health Nursing, Kenyatta University, Nairobi, Kenya
[5] Department of Electrical and Electronic Engineering, Kenyatta University, Nairobi, Kenya
[6] Centre for Virus Research, kenya Medical Research Institute, Nairobi, Kenya
[7] Department of Pharmacology and Clinical Pharmacy, Kenyatta University, Nairobi, Kenya
[8] Department of Biochemistry, Microbiology, and Biotechnology, Kenyatta University, Nairobi, Kenya

Corresponding author
Eric M. Ndombi,
emakuto@gmail.com

## ABSTRACT

**Background:** Post-coronavirus disease 2019 (post-COVID-19) is associated with considerable morbidity and reduced quality of life. However, studies characterizing the post-COVID-19 condition in Kenya are limited. This study aimed to determine the prevalence of post-COVID-19 condition and determine the clinical characteristics, anti-SARS-CoV-2 IgG titers, and concentrations of inflammatory markers of individuals with post-COVID-19 condition in Kenya.

**Methods:** This descriptive cross-sectional study was conducted at the Kenyatta University Health Unit, Kenya. Demographic and clinical data were collected using a questionnaire. The serum levels of anti-SARS-CoV-2 antibodies, interleukin 6 (IL-6), and C-reactive protein (CRP) were quantified by enzyme-linked immunosorbent assays. Independent samples t-test was used to compare the anti-SARS-CoV-2 IgG, IL-6, and CRP levels between the participants with and without post-COVID-19 symptoms. The case definition for post-COVID-19 condition was persistence of acute COVID-19 symptoms or emergence of new symptoms 3 months after COVID-19 diagnosis, symptoms lasting for ≥2 months, and absence of any other etiological basis to explain the symptoms.

**Results:** A total of 189 volunteers were recruited in this study (median age: 21 years, range: 18–71 years; male, 49.2%). Forty participants reported having had at least one COVID-19 positive diagnosis in the past, of which 12 (30%) complained of post-COVID-19 symptoms. Significant differences in the number and duration of symptoms were observed between the individuals with and without post-COVID-19 symptoms (t-statistic = 2.87, p = 0.01; t-statistic = 2.39, p = 0.02, respectively). However, no significant differences in serum levels of anti-SARS-CoV-2 IgG, IL-6, and CRP were observed between the two groups (P = 0.08, 0.9, and 0.28, respectively).

**Conclusion:** These findings suggest that post-COVID-19 condition is a health concern even for a relatively young population in Kenya and globally. This condition requires more attention and well-designed studies to better define it and identify clinical chemistry markers that can be used for its diagnosis.

# INTRODUCTION

Post-coronavirus 2019 condition (post-COVID-19 condition, PCC), also known as long COVID, is recognized worldwide as a healthcare problem that continues to affect the health and quality of life of communities. It is defined by persisting or emerging symptoms after recovery from an infection of severe acute respiratory syndrome coronavirus 2 (SARS-CoV-2). Patients with PCC present with multisystem complaints, including cough, fever, fatigue, dyspnea, brain fog, palpitations, anorexia, anosmia, and ageusia, which persist for many weeks or months. Consequently, the health-related quality of life of the patients is adversely affected (*Smith et al., 2023*). The prevalence of PCC is reported to be between 10% and 20% by the *World Health Organization (2022)*, but some studies report a burden above 50% (*Ogoina, James & Ogoinja, 2021*; *Fernández-de-Las-Peñas et al., 2022*). PCC is commonly believed to affect patients who had severe COVID-19, and existing global research has mainly targeted the hospitalized cases (*Woodrow et al., 2023*; *CDC, 2024*). By the end of 2022, the prevalence of COVID-19 in Africa was approximately 2% of the disease burden worldwide (*Statista, 2022*). In Kenya, the total number of COVID-19 cases was 340,784, with approximately 5,600 COVID-19-related deaths (*Statista, 2022*). Given the relatively lower prevalence, as well as less severity, of COVID-19 in Kenya and the African region, there has been little research focus on PCC in this population. Reports on PCC in Africa are scattered with one study in Nigeria reporting a prevalence of 56.7% among hospitalized patients (*Ogoina, James & Ogoinja, 2021*) and another study in Sub Saharan Africa reporting a prevalence of 9.9% among hospitalized and non-hospitalized patients (*Karuna et al., 2023*). To our knowledge, no study on PCC in Kenya has been published.

The immune system is suggested to influence the development of PCC. *Bichara et al. (2021)* and *Hackenbruch et al. (2023)* observed that anti-SARS-CoV-2 IgG antibodies remained high past 90 days after COVID-19 diagnosis in individuals with more symptoms and severe pulmonary involvement. SARS-CoV-2 has been reported to persist in the body, as indicated by shedding of its nucleic acid in stool 7 months after recovery from acute COVID-19 (*Natarajan et al., 2022*). In addition, SARS-CoV-2 RNA has been detected in various tissues, including the heart, gastrointestinal system, brain, and adrenal glands, up to 230 days after infection (*Stein et al., 2022*). Also, low levels of anti-SARS-CoV-2 IgG antibodies at the acute phase of COVID-19 have been reported to predict persisting symptoms after 6 to 7 months (*Augustin et al., 2021*). The low levels of antibodies in the acute phase correspond to a weak humoral response and could lead to the persistence of

the disease. These findings suggest a possible association between the immune response and the development of PCC symptoms.

In addition, the inflammatory system is considered to play a role in PCC through persistent inflammation, immunosuppression, and catabolism syndrome (*Oronsky et al., 2023*). This condition is characterized by low-grade inflammation across various organs, including the lungs, brain, heart, and gastrointestinal system. Fibroproliferative damage of the lungs following chronic inflammation leads to dyspnea, whereas myocardial inflammation is reported to cause arrhythmia and heart failure (*Castanares-Zapatero et al., 2022*). The potential role of the inflammatory pathway in post-COVID-19 complaints is depicted by an increase in inflammatory markers, such as IL-6, fibrinogen, neutrophils, and C-reactive protein (CRP), in individuals with post-COVID-19 condition (*Maamar et al., 2022*; *Schultheiß et al., 2022*). Therefore, inflammatory markers may potentially be useful as predictive factors for post-COVID-19 complaints.

Owing to the paucity of data on PCC in Kenya, there is a need to address this epidemiological gap and contribute to the definition of this emerging phenomenon. In addition, the mechanisms underlying PCC have not been clearly elucidated, and researchers are endeavoring to unravel why some individuals experience prolonged PCC symptoms. There is also a need to identify possible biomarkers that could be useful in diagnosing or predicting post-COVID-19 condition. The current study focuses on IL-6 and CRP as potential biomarkers for PCC, as IL-6 is centrally involved in COVID-19 pathology as a pro-inflammatory marker (*Batiha et al., 2022*) and CRP is a sensitive marker of inflammation (*Luan, Yin & Yao, 2021*). As part of its 2023–2025 strategic plan, the *World Health Organization (2023)* recommends a broadening of research in local communities to further characterize PCC and understand its impact on people's health. This study aimed at characterizing individuals with PCC in terms of clinical characteristics, anti-SARS-CoV-2 IgG titers, and inflammatory markers.

# MATERIALS AND METHODS

## Study design and setting

This descriptive cross-sectional study was conducted at the Kenyatta University Health Unit between April and July 2023. Kenyatta University, one of the top three public universities in Kenya in terms of population and diversity, is located in the capital city of Kenya, Nairobi, and comprises students and staff from various social, economic, religious, and geographic backgrounds. Of note, Kenya has a youthful population and according to a recent demographics survey, the median age in Kenya is 19.9 years, with 75% of the population aged 0–34 years (*National Council for Population & Development, 2021*). As such, a study sample from Kenyatta University may be a representative of the Kenyan population, as majority of students in the university are between 18 years to 25 years old. The study was granted ethical approval by the Kenyatta University Ethics Review Committee (Reference number PKU/2379/11516). Written informed consent was obtained from the participants after explaining all the study details, including the procedures, discomfort, risks, and benefits.
## Participants and eligibility criteria

Healthy volunteers from Kenyatta University Community, including students and staff, were eligible for this study regardless of their COVID-19 history and comorbidities. A call for participation was advertised through posters placed at different locations across the university. In addition, the posters were shared in WhatsApp groups of students and staff. All the participants who responded to the call for participation were consented for enrollment into the study. The inclusion criteria were (1) a student or staff at Kenyatta University, (2) aged ≥18 years, and (3) absence of acute health complaints unrelated to COVID-19 at enrollment. Using Fisher's formula, $n = (Z^2PQ)/E^2$, where $n$ = the desired sample size, Z = 95% confidence interval (standard value of 1.96), $P$ = estimated prevalence of PCC in Sub Saharan Africa (9.9%) (*Karuna et al., 2023*), E = probable random error (5%), and Q = 1–P, a minimum sample size of 137 participants was required in this study.

## Questionnaire data and PCC symptoms

A structured questionnaire was used to collect the participants' clinical and demographic data. The clinical data were self-reported by the participants and included the previous diagnosis of COVID-19, number and duration of symptoms, presence and type of persisting COVID-19 symptoms, and presence and type of new onset symptoms. COVID-19 vaccination-related data were also collected by confirming the individual vaccination status of participants in a portal maintained by the Ministry of Health, Kenya. A case of PCC in the present study was defined as persistence of acute COVID-19 symptoms or emergence of new symptoms 3 months following COVID-19 diagnosis, symptoms lasting for ≥2 months, and absence of any other etiological basis to explain the symptoms, according to the *World Health Organization (2022)*.

## Sample collection and measurement of anti-SARS-CoV-2 IgG, IL-6, and CRP

Blood samples were collected from the participants and centrifuged immediately to obtain serum, which was stored at −20 °C until testing for anti-SARS-CoV-2 IgG, IL-6, and CRP levels. The level of anti-SARS-CoV-2 IgG in the serum was quantified using human SARS-CoV-2 Spike (Trimer) IgG enzyme-linked immunosorbent assay (ELISA) (Thermo Fisher Scientific, Waltham, MA, USA), as detailed by the manufacturer. The assay was based on sandwich ELISA in which wells were precoated with a trimerized spike protein. The serum was diluted 1,000-fold and 10 μL of the diluted sample added to the wells. Biotin-conjugated IgG antibodies were used to detect the captured anti-SARS-CoV-2 IgG antibodies, and the absorbances were read at a wavelength of 450 nm using the RT-2100C spectrophotometer (Rayto Life and Analytical Sciences, Guangdong, China).

CRP was assayed using Human CRP ELISA kit (FineTest, Wuhan Fine Biotech Co., Ltd, Wuhan, China), whereas IL-6 was measured *via* Human IL-6 ELISA kit (FineTest, Wuhan Fine Biotech Co., Ltd, Wuhan, China), according to the manufacturer's instructions. The CRP and IL-6 assays were based on sandwich ELISA in which antibodies against CRP and IL-6, respectively, were precoated on the wells of 96-well plates. For the CRP assay, 100 μL of a 20-fold pre-diluted sample was added to each well and biotin-conjugated

anti-CRP used as the detection antibody. On the other hand, for the IL-6 assay, 100 µL of a 2-fold pre-diluted sample was added to each well, and anti-IL-6 antibodies were used to detect the IL-6. The absorbances were read at 450 nm using the RT-2100C spectrophotometer.

## Statistical analyses

Descriptive statistics were used to analyze the characteristics of the participants, including age, sex, COVID-19 diagnosis and symptoms, duration of persistent symptoms, and COVID-19 vaccination status. The proportion of participants with post-COVID-19 complaints was determined using frequencies. The number and duration of symptoms were compared using the independent samples $t$-test. Relationships between the number of COVID-19 symptoms and anti-SARS-CoV-2 IgG titers were determined using the Pearson correlation test. The independent $t$-test was used to compare anti-SARS-CoV-2 IgG, IL-6, and CRP levels between individuals with and without PCC symptoms. The difference in vaccination status between the two groups was determined using Chi-square test. Statistical significance was denoted by a $p$-value $< 0.05$. Statistical analysis was performed using SPSS Version 18 (IBM Corp, Armonk, NY, USA).

## RESULTS

### Participants' demographic and clinical characteristics

In total, 189 volunteers participated in this study (median age: 21 years, range: 18–71 years; male, 50.8%). A total of 40 individuals indicated they had COVID-19, out of which 12 (30%) presented with post-COVID-19 symptoms (persistence of symptoms: $n = 6$, emergence of new symptoms: $n = 7$). One of the participants had both persisting and new symptoms. The study flowchart is shown in Fig. 1.

The clinico-demographic characteristics of the participants with COVID-19 history are presented in Table 1. Among the 40 participants who reported having had COVID-19 in the past, 70% were aged 20–29 years and 52.5% were female. All cases of COVID-19 were mild or moderate. The COVID-19 vaccination rate was 65%. We observed that the last COVID-19 diagnosis for 77.5% of the participants was in the last 1–2 years. Among the comorbidities, asthma was reported by two participants.

Among the 12 participants with post-COVID-19 symptoms, the most common persisting symptoms were cough ($n = 3$, 50%), sore throat ($n = 2$, 33.3%), and runny/stuffy nose ($n = 2$, 33.3%), whereas the most common new symptoms included fatigue ($n = 3$, 42.9%) and loss of smell/taste ($n = 3$, 42.9%) (Fig. 2). The less frequent symptoms were diarrhea ($n = 1$, 16.7%) and fatigue ($n = 1$, 16.7%) among persisting symptoms and dizziness ($n = 1$, 14.3%), altered menstrual cycle ($n = 1$, 14.3%), earache ($n = 1$, 14.3%), shortness of breath ($n = 1$, 14.3%), depression and anxiety ($n = 1$, 14.3%), and loss of appetite ($n = 1$, 14.3%) among new symptoms. Only fatigue was identified as both a persisting and new symptom. For all the participants, the post-COVID-19 symptoms had lasted for $\geq 1$ year. Of the 12 participants, two had asthma.

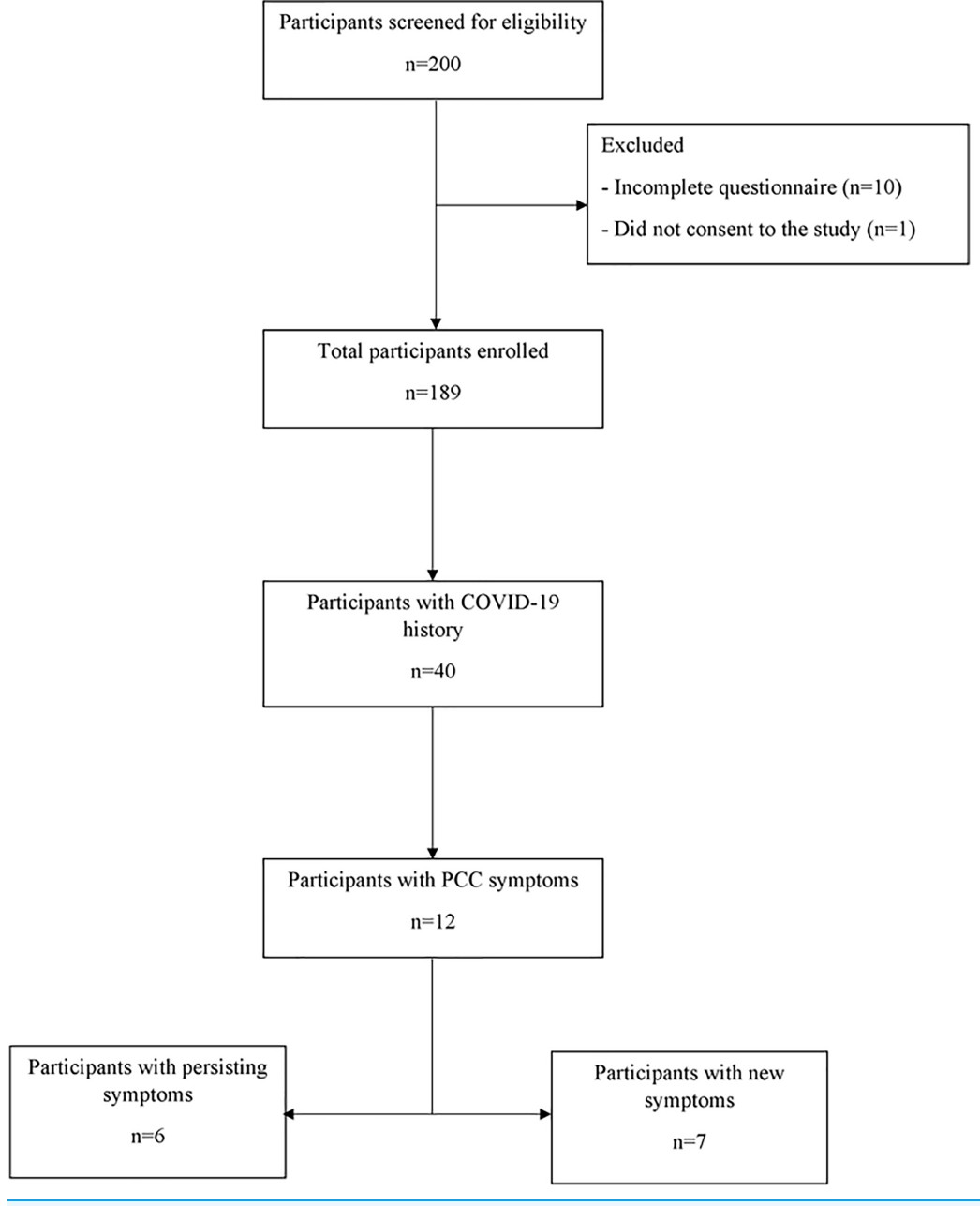

**Figure 1 Flow chart showing total number of participants and subsets based on their clinical characteristics and reported COVID-19 history.** Among the participants with PCC symptoms, one had both persisting and new-onset symptoms. COVID-19, coronavirus disease 2019; PCC, post-COVID-19 condition.

## A comparison of clinical characteristics between individuals with and without PCC

The mean number of symptoms among individuals with post-COVID-19 symptoms was >7, whereas that among those without post-COVID-19 symptoms was <6. A significant difference in the number of symptoms was observed between the individuals with and without post-COVID-19 symptoms ($t$-statistic = 2.87, $p$ = 0.01) (Table 2). In addition, the

**Table 1 Characteristics of the participants with COVID-19 history.**

| | Variables | Frequency (N) | Percentages (%) |
|---|---|---|---|
| Sex | Male | 19 | 47.5 |
| | Female | 21 | 52.5 |
| Age (years) | 18–19 | 4 | 10 |
| | 20–29 | 28 | 70 |
| | 30–39 | 3 | 7.5 |
| | 40–49 | 4 | 10 |
| | ≥50 | 1 | 2.5 |
| Vaccination status | Vaccinated | 26 | 65 |
| | Not vaccinated | 14 | 35 |
| Current occupation | Teaching staff | 2 | 5 |
| | Non-teaching staff | 3 | 7.5 |
| | Students | 35 | 87.5 |
| Duration from last COVID-19 diagnosis | <6 months | 1 | 2.5 |
| | 6 months to <1 year | 4 | 10 |
| | 1–2 years | 31 | 77.5 |
| | >2 years | 4 | 10 |
| Comorbidities | Diabetes | 0 | 0 |
| | Hypertension | 0 | 0 |
| | HIV | 0 | 0 |
| | Asthma | 2 | 5 |
| | Cancer | 0 | 0 |
| | Autoimmune disease | 0 | 0 |
| COVID-19 severity | Mild/moderate | 40 | 100 |
| | Severe | 0 | 0 |

mean duration of recovery differed significantly between those with and without post-COVID-19 symptoms (15.17 days *vs.* 10 days, *t*-statistic = −2.35, *p* = 0.02) (Table 2). In both groups, a positive correlation was observed between the number of symptoms and duration of recovery, but the correlations were relatively weak and not statistically significant (With post-COVID-19 symptoms: *r* = 0.12, *p* = 0.7; Without post-COVID-19 symptoms: *r* = 0.17, *p* = 0.37). The rate of vaccination was not significantly different between the individuals with and without post-COVID-19 symptoms (with post-COVID-19 symptoms: 58.3% [7/12]; without post-COVID-19 symptoms: 64.3% [18/28], *p* = 0.72) (Table 2).

## A comparison of anti-SARS-CoV-2 IgG, CRP, and IL-6 between individuals with and without PCC

The group with post-COVID-19 symptoms had a lower mean of anti-SARS-CoV-2 IgG levels compared to the group without post-COVID-19 symptoms, although both groups were not statistically different ($1.7 \times 10^8$ *vs.* $2.1 \times 10^8$ Units/mL, *t*-statistic = −1.7, *p* = 0.08) (Table 2). Also, the number of COVID-19 symptoms did not correlate significantly with

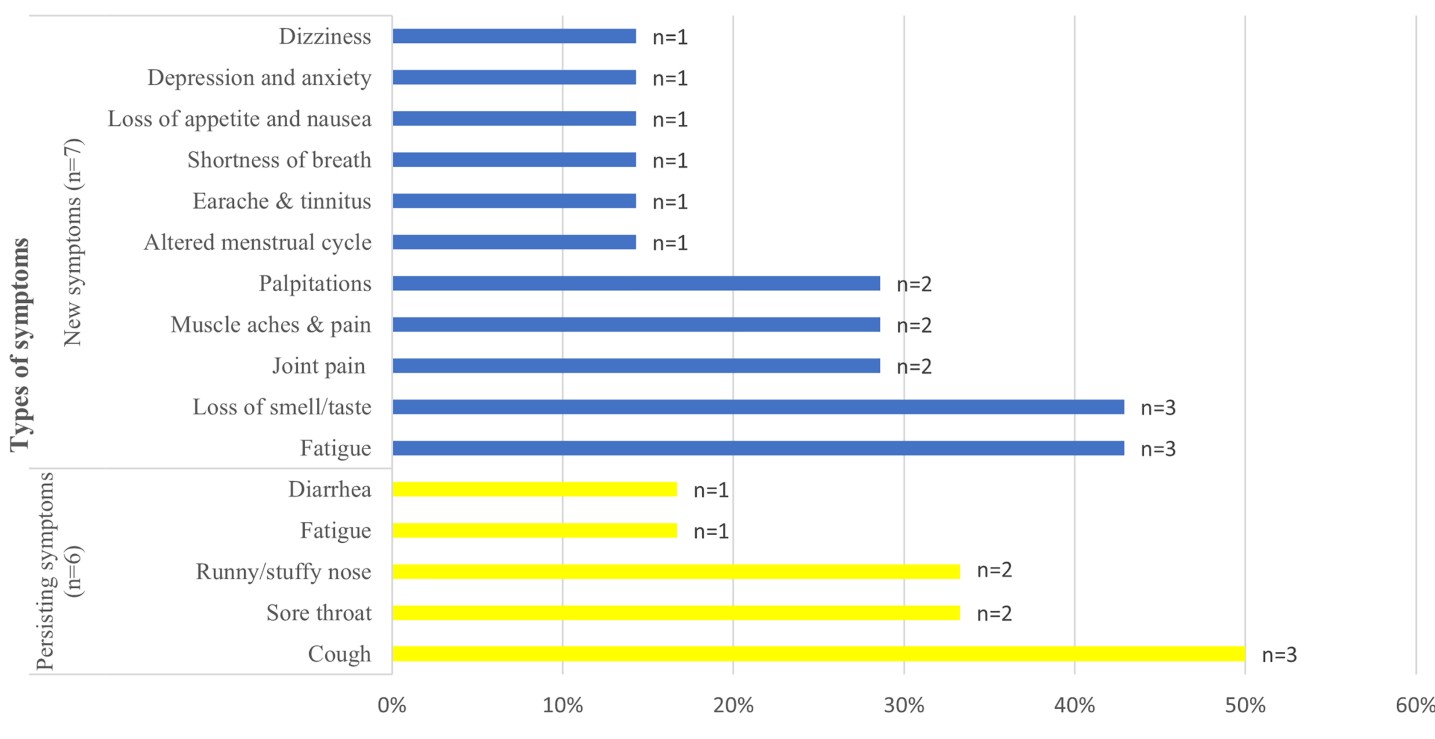

**Figure 2 New and persisting COVID-19 symptoms, their frequencies and percentages.** Frequencies of persisting symptoms of COVID-19 and new symptoms after acute COVID-19. Types of symptoms are provided on the y-axis and percentage of participants with a particular symptom on the x-axis. The data label at the end of each bar graph indicates the absolute number of participants reporting a particular symptom. Blue color indicates the new symptoms and yellow color indicates the persisting symptoms. COVID-19, coronavirus disease 2019.

**Table 2 Comparison of anti-SARS-CoV-2 IgG, IL-6, and CRP between participants with and without PCC symptoms.**

|  | Variables | With PCC symptoms | Without PCC symptom | *P*-value |
|---|---|---|---|---|
| Number of symptoms (n) |  | >7 | <6 | 0.01[a] |
| Duration of symptoms (days) |  | 15.17 | 10 | 0.02[a] |
| Vaccination status (n) | Vaccinated | 7 (58.30%) | 18 (64.30%) | 0.72[b] |
|  | Not vaccinated | 5 (41.70%) | 10 (35.70%) |  |
| IL-6 (pg/ml) |  | 11.40 | 8.10 | 0.90[a] |
| CRP (pg/ml) |  | 45,856.38 | 42,427 | 0.28[a] |
| Anti-SARS-COV-2 IgG (Units/mL) |  | $1.7 \times 10^8$ | $2.1 \times 10^8$ | 0.08[a] |

Notes:
[a] Independent sample t-test,
[b] Chi square test.
IgG, immunoglobulin gamma; IL-6, interleukin 6; PCC, post-COVID-19 condition; SARS-CoV-2, severe acute respiratory syndrome coronavirus 2.

the IgG levels in individuals with and without post-COVID-19 symptoms ($r = -0.15$, $p = 0.63$; $r = 0.25$, $p = 0.2$, respectively). Regarding CRP, there was no significant difference between the groups with and without post-COVID-19 symptoms ($t$-statistic: 1.09, $p = 0.28$). Also, IL-6 levels did not significantly differ between the two groups ($t$-statistic: 0.13, $p = 0.9$) (Table 2).

## DISCUSSION

This cross-sectional study reported a prevalence of 30% for PCC, with six and seven individuals complaining of persistent and new-onset symptoms, respectively. Cough, sore throat, and runny/stuffy nose were the most common persisting symptoms, whereas loss of smell/taste and fatigue were the most frequent new symptoms. The number of COVID-19 symptoms and mean duration of recovery differed significantly between those with and without PCC. However, no significant differences in anti-SARS-CoV-2 IgG, IL-6, and CRP were observed between the individuals with and without PCC. Also, the antibody levels did not correlate with the number of COVID-19 symptoms.

As in the present study, PCC is characterized by either persisting or new symptoms. Fatigue, sore throat, and cough, which were observed in this study, are some of the classic symptoms of COVID-19 and have been reported to persist in various studies (*Durstenfeld et al., 2023*; *Yang et al., 2022*; *Förster et al., 2022*). A persistent infection may indicate an incomplete immune response, and the underlying pathophysiology is likely similar to that of acute COVID-19 (*Jacobs, 2021*). It is postulated that the presence of residual viral particles in the body due to delayed clearance triggers different immune pathways, including inflammation and direct viral killing. On the other hand, new onset symptoms could point to delayed immune activity against the virus or reactivation of the virus. Some of the reported symptoms such as depression, anxiety, and fatigue are not specific to SARS-CoV-2 and could be due to the social effects of the COVID-19 pandemic (*Durstenfeld et al., 2023*). Also, two participants with PCC had asthma, which is characterized by chronic inflammation of the airways and presents with symptoms such as cough, shortness of breath, and runny nose (*Kwok et al., 2023*). Therefore, for these participants, it may be difficult to distinguish PCC from asthma exacerbation (*Kwok et al., 2023*). With more research on the spectrum of diseases associated with SARS-CoV-2, viral shedding, immune response, and long-term pathophysiology of SARS-CoV-2, it will be possible to gain a robust understanding of PCC.

Multisystem inflammation is postulated as a potential mechanism that can explain the various specific symptoms of PCC (*Durstenfeld et al., 2023*). The respiratory symptoms, such as cough, sore throat, runny nose, and loss of smell/taste, are thought to arise from inflammatory damage of the lungs and tissues along the respiratory tract (*Tandon et al., 2024*). On the other hand, palpitations could be due to myocardial inflammation, which alters the function of the heart muscles (*Castanares-Zapatero et al., 2022*). In addition, fatigue, as well as joint and muscle pain, are musculoskeletal symptoms of PCC, which could be related to inflammatory processes triggered by the direct invasion of the muscles and joints by SARS-CoV-2 (*Zadeh, Wilson & Agrawal, 2023*). Inflammatory cytokines, such as IL-1 and IL-6, have also been reported to alter the balance of cortisol and adrenocorticotropic hormone, leading to fibromyalgia and chronic fatigue (*Zadeh, Wilson & Agrawal, 2023*). However, the role of the inflammatory pathway in the clinical manifestations in the present study is uncertain, given that the concentrations of inflammatory markers, IL-6 and CRP, were similar in all individuals, irrespective of post-COVID-19 symptoms.

The absence of significant difference in IL-6 between individuals with and without post-COVID-19 symptoms contrasts the results of a recent meta-analysis (*Yin et al., 2023*), which showed an elevation of IL-6 in individuals presenting with post-COVID-19 symptoms compared to healthy individuals. Another study by *Schultheiß et al. (2022)* also reported an upregulation of IL-6 among patients with post-COVID-19 complaints, 8 months after recovery from acute COVID-19. Similar to this study, *Queiroz et al. (2022)* reported that IL-6 was not significantly elevated in people with post-COVID-19 symptoms. However, high levels of other cytokines, including IL-17 and IL-2, and low levels of IL-4 and IL-10 comprised the cytokine profile of PCC (*Queiroz et al., 2022*). Further research is required to clarify the role of IL-6 in PCC.

Similarly, CRP, which is a sensitive marker of inflammation, was also assayed, but it did not differ between individuals with and without PCC. The production of CRP is related to the release of pro-inflammatory cytokines (*Abdullah et al., 2023*), and many recent studies have reported elevated levels of CRP in individuals with PCC (*Lai et al., 2023*; *Giridharan et al., 2023*; *Xuereb et al., 2023*). In the study by *Maamar et al. (2022)* abnormally high levels of CRP were observed in participants complaining of post-COVID-19 symptoms, suggesting low-grade inflammation. CRP is a known marker of inflammation and the lack of significant difference in the present study necessitates more research in the future.

Of note, age has been reported as a key risk factor for PCC (*Sudre et al., 2021*; *Tsampasian et al., 2023*), and the lack of significant differences in the inflammatory markers could be due to the study of a homogenous population in terms of age. Seventy percent of the participants with positive COVID-19 history in the present study were aged 20–29 years. This age group is likely to have mild COVID-19 and thus systemic derangement and inflammation may be less severe compared to that of older people aged above 65 years (*Hu et al., 2023*). In addition, the risk of inflammation increases with age, and older people are reported to have a pro-inflammatory state (*Müller & Di Benedetto, 2023*). Thus, older adults are likely to have a marked increase in IL-6 and CRP, compared to young people, following inflammatory signaling in PCC.

Anti-SARS-CoV-2 IgG levels are markers of the body's immune response against the SARS-CoV-2 virus. Given the hypothesis that persistent symptoms imply an incomplete immune response, patients with PCC may have lower titers of anti-SARS-CoV-2 IgG (*Jacobs, 2021*). Despite the absence of statistical significance, individuals with PCC symptoms in the present study had lower anti-SARS-CoV-2 IgG titers than those without PCC symptoms. A study by *García-Abellán et al. (2021)* also observed lower titers of anti-SARS-CoV-2 IgG in individuals presenting with PCC symptoms at 6 months. The lack of a potent immune response, including insufficient production of neutralizing antibodies, may impair the control of viral replication, leading to long-term sequelae of COVID-19 (*García-Abellán et al., 2021*). This may also be associated with a higher disease severity. On the other hand, *Peghin et al. (2021)* reported persistently high anti-SARS-CoV-2 IgG titers in individuals with lingering symptoms 6 months after acute COVID-19. The levels of anti-SARS-CoV-2 IgG are expected to reduce in patients recovering from acute COVID-19 following the clearance of the virus from the body. However, if the virus remains in the body, anti-SARS-CoV-2 IgG titers are expected to be high.

Vaccination boosts the natural immunity for enhanced protection against the virus. The COVID-19 vaccination rate in the present study was not significantly different between individuals with and without PCC symptoms. Similarly, a study by *Kim et al. (2024)* reported no difference in the occurrence of PCC symptoms irrespective of the vaccination status or doses of the COVID-19 vaccine. It is also reported that vaccination may have minimal efficacy against pre-existing PCC complaints (*Notarte et al., 2022*). In contrast, a recent study has revealed that COVID-19 vaccines are effective in preventing PCC symptoms (*Català et al., 2024*). The authors suggest that COVID-19 vaccination protected against PCC symptoms by preventing severe disease. Even a single dose of COVID-19 vaccine is effective against PCC, although two doses of the vaccine are associated with a lower risk of persistent symptoms (*Antonelli et al., 2022*). Given the discrepant results regarding the effectiveness of vaccination against PCC, a prospective study with a larger sample size and long follow-up should be performed to measure the efficacy of COVID-19 vaccines for long-term effects of COVID-19.

Some limitations of this study should be acknowledged. First, as the number of participants with post-COVID-19 complaints was low, which was expected given the young demographic that formed the bulk of our participants and the fact that none of them reported having suffered severe Covid-19 disease at the time of infection. We were therefore not able to assess the relationship between anti-SARS-CoV-2 IgG titers and specific PCC symptoms. Second, COVID-19 history and PCC symptoms were self-reported, which could bias our results. However, we consider this approach somewhat suitable for our study, which enrolled participants from the general community who were not hospitalized for COVID-19. The usefulness of self-reported data in the diagnosis of COVID-19 and characterization of emerging conditions such as PCC has been emphasized (*Mockler et al., 2022*; *Huijts et al., 2023*). Third, the analysis could not be stratified according to age, as majority of the participants were in their 20 s. Therefore, a study that considers a broad age range in recruiting participants can generate richer data about the possible influence of age on occurrence of PCC. This study nonetheless contributes to the understanding of PCC and highlights the need to include even those individuals generally considered not to be at risk of this complex and evolving condition.

## CONCLUSIONS

The present study suggests that PCC symptoms could be prevalent in the general population, including among healthy young adults who suffered mild or moderate COVID-19 who comprised the bulk of participants in this study. While the major concerns of COVID-19 at the height of the pandemic in terms of severe disease and fatalities have waned, PCC warrants attention of the health stakeholders. The levels of anti-SARS-CoV-2 IgG, CRP, and IL-6 were not significantly different between individuals with PCC compared to those without PCC in this study. It is recommended that well-designed prospective studies be carried out to better define PCC and identify clinical chemistry markers that can be used for its diagnosis.

## ACKNOWLEDGEMENTS

The authors would like to thank the staff of the Directorate of Kenyatta University Health Services, Nairobi, Kenya, for their assistance in guiding participants and in sample collection and storage. Special appreciation also goes to the participants for their willingness to take part in the study.

### Funding

This work was supported by the National Research Foundation (NRF) South Africa (COV19200617532972). The funders had no role in study design, data collection and analysis, decision to publish, or preparation of the manuscript.

### Grant Disclosures

The following grant information was disclosed by the authors:
National Research Foundation (NRF) South Africa: COV19200617532972.

### Competing Interests

The authors declare that they have no competing interests.

### Author Contributions

- Martin Theuri performed the experiments, analyzed the data, prepared figures and/or tables, authored or reviewed drafts of the article, and approved the final draft.
- Eric M. Ndombi conceived and designed the experiments, performed the experiments, analyzed the data, prepared figures and/or tables, authored or reviewed drafts of the article, and approved the final draft.
- Peris Thamaini conceived and designed the experiments, prepared figures and/or tables, and approved the final draft.
- James Opiyo Ogutu conceived and designed the experiments, prepared figures and/or tables, and approved the final draft.
- Lister Onsongo conceived and designed the experiments, prepared figures and/or tables, and approved the final draft.
- June K. Madete conceived and designed the experiments, prepared figures and/or tables, and approved the final draft.
- Victor Ofula conceived and designed the experiments, prepared figures and/or tables, and approved the final draft.
- Samuel Gitau conceived and designed the experiments, prepared figures and/or tables, and approved the final draft.
- Gladys Mwangi conceived and designed the experiments, prepared figures and/or tables, authored or reviewed drafts of the article, and approved the final draft.
- Paul Okemo conceived and designed the experiments, prepared figures and/or tables, and approved the final draft.
## Human Ethics

The following information was supplied relating to ethical approvals (*i.e.*, approving body and any reference numbers):

Kenyatta University Ethical Review Committee granted ethical approval to carry out the study within the university (Ethical Application Reference number PKU/2379/11516).

## Data Availability

The raw data is available in the Supplemental File.

## Supplemental Information

Supplemental information for this article can be found online at http://dx.doi.org/10.7717/peerj.17723#supplemental-information.

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
