# Peer review of "Clinical characteristics, anti-SARS-CoV-2 IgG titers, and inflammatory markers in individuals with post-COVID-19 condition in Kenya: a cross-sectional study"

_PeerJ, doi:10.7717/peerj.17723_

## Round 0.1 · original submission · Major Revisions

This is a relevant study but as both reviewers have pointed out the study's design and sample size is unsatisfactory. Also the discussion lacks weight and needs better construction and thought. Please address the critiques and resubmit.

Reviewer 1 ·

Basic reporting

The paper is clear and unambiguous in what it wants to address. The English is good, but there were just some few revisions and errors that need to be corrected (see Additional Comments).

The Introduction lacks discussion of the status of COVID-19 & PCC in Africa & Kenya where the setting of the study is.

Experimental design

The research question is well defined and relevant; however some concerns and questions need to be addressed.

• Justify the use of the Kenyatta University Community as the source of study subjects. This can be a limitation of the study because of the particular characteristics of the participants which may not be representative of the population.

• Was sample size calculation done for the study?

• How did the researchers ascertain the diagnosis of COVID-19 among participants and their vaccination status? Are there supporting documents or records given? Or are they just self-reported?

• The specific comorbidities and severity of COVID-19 were not included in the data collected, which are important factors in explaining the observed immune response from participants.

• How did the researchers know that the participants' IgG levels were due to COVID-19 infection and not due to their vaccination?

Validity of the findings

The Conclusion that "PCC symptoms are prevalent in the general population" is an overstatement in the context of this study.

Additional comments

• Lines 73-76 came from one reference only, so no need to repeat citation.
• Line 105: should be "between April and July 2023"
• Line 115: Regarding the Inclusion criteria, mention that participants were recruited regardless of their history of COVID-19 infection and comorbidities.
• Line 136: "Ig-G" should be "IgG"
• Line 138: The type and manufacturer of the spectrophotometer used were not mentioned
• Figure 1 should include the # of individuals screened, # of successfully enrolled and reasons for the exclusion of some, if any. It should also include a footnote explaining that one participant had persisting and new symptoms.
• Line 181: Kindly provide details about the asthma of the two participants? Is their asthma controlled? Are they on medications?
• In Table 1: Age ≤ 19 should just be “18-19”
• Duration of symptoms is better expressed as number of days, rather than number of weeks.
• In Table 2, be consistent with the # of decimal places used in the values.

Reviewer 2 ·

Basic reporting

The study has no impact on Long COVID, It has failed to see any major symptoms of Post COVID conditions. Authors also didnt mentioned the history or when they developed COVID (year) and whether they had moderate or Severe COVID when they are infected. And also most of the laboratory findings are insignificant with and without postCOVID condition symptoms.

Experimental design

Its a cross sectional study majorly focused on the students and staff. Need more valuable questionaries for post COVID symptoms.

Validity of the findings

There is no novelty in the study and cant claim IL-6 or CRP as the biomarkers as they are evaluated only to very small populations.

---

## Round 0.2 · accepted · Accept

We appreciate that you took the time to address all the reviewer comments and bring the manuscript to its current form, where it is ready for publication.

Reviewer 1 ·

Basic reporting

No Comment

Experimental design

No Comment

Validity of the findings

No Comment

Additional comments

No Comment

Reviewer 2 ·

Basic reporting

As the Author, commented and changed all the sections of the queries raised by the Reviewers, It seems the article is well-rewritten now and accepted for publication.

Experimental design

No comments

Validity of the findings

No comments

Additional comments

Adding and re-editing all those comments increased the weightage to be published.